# Inner-shell clock transition in atomic thulium with a small blackbody radiation shift

A. Golovizin[1], E. Fedorova[1,2], D. Tregubov[1], D. Sukachev [1,3], K. Khabarova[1,2], V. Sorokin[1] & N. Kolachevsky[1,2]

One of the key systematic effects limiting the performance of state-of-the-art optical clocks is the blackbody radiation (BBR) shift. Here, we demonstrate unusually low sensitivity of a 1.14 µm inner-shell clock transition in neutral Tm atoms to BBR. By direct polarizability measurements, we infer a differential polarizability of the clock levels of −0.063(30) atomic units corresponding to a fractional frequency BBR shift of only $2.3(1.1) \times 10^{-18}$ at room temperature. This amount is several orders of magnitude smaller than that of the best optical clocks using neutral atoms (Sr, Yb, Hg) and is competitive with that of ion optical clocks (Al$^+$, Lu$^+$). Our results allow the development of lanthanide-based optical clocks with a relative uncertainty at the $10^{-17}$ level.

[1] P.N. Lebedev Physical Institute, Leninsky prospekt 53, Moscow 119991, Russia. [2] Russian Quantum Center, Business-Center "Ural", 100A Novaya St. SkolkovoMoscow 143025, Russia. [3] Physics Department of Harvard University, 17 Oxford Str., Cambridge, MA 02138, USA. Correspondence and requests for materials should be addressed to N.K. (email: kolachevsky@lebedev.ru)

The unprecedented performance of state-of-the-art optical atomic clocks[1-3], together with the increasing number of characterized clock transitions in different atomic and ionic species, has pushed the frontiers of physics enabling sensitive tests of special relativity[4,5], the search for variations of fundamental constants[6,7] and dark matter[8,9], atom-based quantum simulations[10], and others as summarized in reviews[11-13]. Moreover, the development of robust, compact, and transportable optical clocks[14], overcoming in accuracy the best microwave frequency standards, promises big progress in precision spectroscopy, navigation, and geodesy[15,16].

The fractional frequency uncertainty of optical clocks reaching a low level of $10^{-18}$ was successfully demonstrated for Sr[17] and Yb[18] optical lattice clocks and for Yb[+19] and Al[+20] single-ion optical clocks. One of the current limitations is the impact of electric fields, such as surrounding blackbody radiation (BBR), trapping fields, and collisions. To extend the current limit in frequency stability and accuracy, there is a continuous search for new species, particularly with reduced sensitivity to external electric fields. Among the promising candidates are single Lu$^+$ ions[21], highly charged ions[22-24], and $^{229}$Th with an isomeric nuclear transition[25,26].

Lanthanides with a submerged electronic $f$-shell possess naturally suppressed sensitivity to external electric fields for inner-shell $f-f$ transitions because of strong shielding from the closed $5s^2$ and $6s^2$ shells. As early as 1984, E. Alexandrov and coauthors demonstrated unusually low spectral broadening of the 1.14 μm inner-shell transition in atomic Tm under collisions with buffer He gas[27]. In 2004, the strong shielding effect for Tm–He collisions was confirmed in ref. [28]. Similar effects were also observed for some transition elements with nonzero orbital angular momentum[29]. For lanthanide ions doped in solids, the shielding effect reduces inhomogeneous broadening of the inner-shell $f-f$ transitions, which allows, for example, ensemble-based solid-state quantum memory[30] and an integrated single-photon source in the telecom wavelength range[31]. However, the possibility to use such transitions for optical clocks was studied only theoretically[32-34].

In this Article, we report on precision spectroscopy of the inner-shell clock transition $|J = 7/2, F = 4, m_F = 0\rangle \rightarrow |J = 5/2, F = 3, m_F = 0\rangle$ between the fine-structure components of the ground electronic state in the single stable isotope $^{169}$Tm (the nuclear spin equals $I = 1/2$) at a wavelength of 1.14 μm with the natural linewidth of $\gamma = 1.2$ Hz (here, $J$ and $F$ stand for the electronic and total momentum quantum numbers, respectively, and $m_F$ is the magnetic quantum number). Spectroscopy of the $|m_F = 0\rangle \rightarrow |m_F = 0\rangle$ clock transition leads to zero first-order Zeeman and magnetic dipole–dipole interaction shifts[34]. We experimentally determined the magic wavelength of the optical lattice near 813 nm, recorded the Fourier-limited spectral linewidth of the clock transition of 10 Hz and characterized its sensitivity to electric and magnetic fields. Accurate measurement of the differential dynamic polarizabilities of clock levels in the near-infrared spectral region (810–860 and 1064 nm) allowed an estimate for the BBR frequency shift of the clock transition, which was found to be a few orders of magnitude smaller than that of other characterized clock transitions in neutral atoms.

## Results

**Experimental setup**. Thulium atoms are laser cooled in a two-stage magneto-optical trap (see Fig. 1 and the detailed description in ref. [35]). For precision spectroscopy, atoms at a temperature of ~10 μK are loaded in a vertical optical lattice formed inside an enhancement cavity (finesse of 20) with intracavity polarization filtering. This configuration allows for 6 W of circulating power in the cavity from a 1 W Ti:sapphire laser between 810 and 860 nm. The beam waist radius can be varied between 80 and 160 μm by controlling the distance between cavity elements.

We prepare atoms in lower clock state $|F = 4, m_F = 0\rangle$ by driving simultaneously $|F = 4\rangle \rightarrow |F = 4\rangle$ (pump) and $|F = 3\rangle \rightarrow |F = 4\rangle$ (repump) transitions with 5 ms-long π-polarized laser

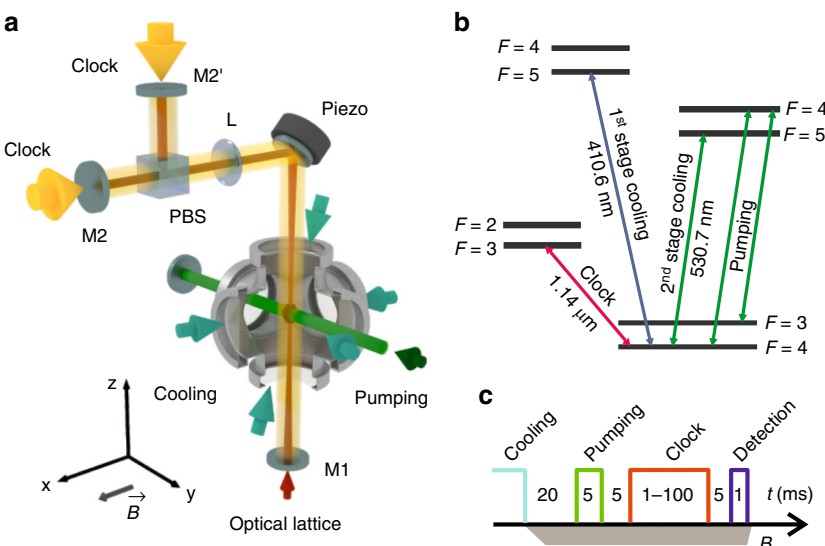

**Fig. 1** Experimental details. **a** Schematic of the setup: the enhancement cavity forming the vertical optical lattice is built in a Γ-shape configuration with an intracavity polarizing beam splitter (PBS). For different polarizations, the two output couplers M2 and M2' are used. The clock laser radiation at 1.14 μm is fed collinearly with the cavity axis through one of the cavity outcouplers with corresponding polarization; the pumping and repumping laser beams form standing waves along the y-axis; and the initial position of the Tm magneto-optical trap is adjusted to the cavity waist. M1: cavity incoupler, L: lens. **b** Relevant Tm level scheme. **c** Pulse sequence for spectroscopy of the clock transition. Two-stage laser cooling is followed by pumping to the $|F = 4, m_F = 0\rangle$ ground state using 530.7 nm radiation. The optical lattice is continuously on. After excitation of the clock transition, one measures the fluorescence of atoms remaining in the ground state by a 1 ms pulse (detection) at 410.6 nm (the same transition as for the first-stage cooling). The bias magnetic field B is denoted by a gray bar on the bottom

pulses at 530.7 nm in the presence of a bias magnetic field of $B_x \sim 0.1$ G. After the optical pumping cycle, $10^5$ atoms are trapped in the lattice, with more than 80% in the target state. To interrogate the clock transition, we use radiation of a semi-conductor 1.14 μm laser stabilized to a high-finesse ultralow-expansion glass (ULE) cavity, providing a relative frequency instability of smaller than $10^{-14}$ over 1–100 s integration time[36]. The linear frequency drift of 29 mHz s$^{-1}$ is compensated using an acousto-optical modulator. After compensation, the clock laser can be used as a stable frequency reference for studying frequency shifts of the clock transition.

**Magic wavelength determination.** The important step toward high-resolution spectroscopy of the clock transition is the determination of the magic wavelength of the optical lattice when the dynamic polarizabilities of the clock levels become equal[37]. We numerically calculated the dynamic polarizabilities of the clock levels using time-dependent second-order perturbation theory and transition data obtained using the COWAN package[38] (Model 1 is described in Methods and in ref. [34]), and we predicted the existence of the magic wavelength at 811.2 nm (for the collinear magnetic field **B** and the lattice field polarization **ε**) near a narrow transition from the upper $J = 5/2$ clock level at 809.5 nm. The calculated differential polarizability is shown in Fig. 2 with the red solid line. This wavelength region is readily accessible by Ti:sapphire or powerful semiconductor lasers.

Using the approach described in ref. [39], we experimentally searched for the magic wavelength for the clock transition in the spectral region of 810–815 nm. The transition frequency shift $\Delta\nu$ as a function of optical lattice power $P$ was measured at different lattice wavelengths (see Methods). The differential dynamic polarizability $\Delta\alpha$ between the clock levels was calculated using the expression

$$\Delta\alpha = -\frac{hcw^2}{16a_0^3\eta}\frac{\Delta\nu}{P}. \quad (1)$$

In the paper, we use atomic units for polarizability, namely, 1 a.u. $= 4\pi\varepsilon_0 a_0^3$. Here, $h$ is the Plank constant, $c$ is the speed of light, $a_0$ is the Bohr radius, and $w = 126.0(2.5)$ μm is the lattice beam radius as calculated from the enhancement cavity geometry.

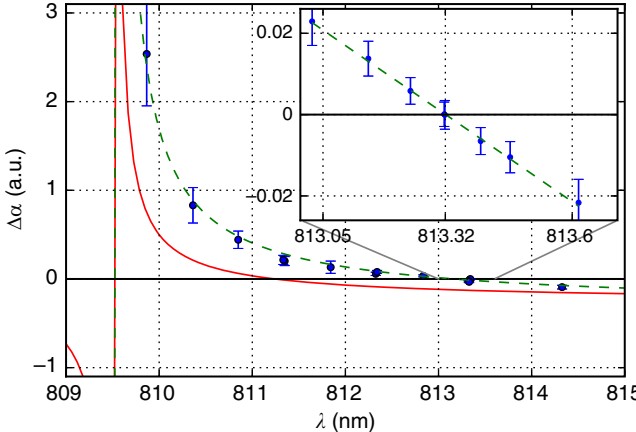

**Fig. 2** Magic wavelength determination. Calculated (Model 1, red solid curve), measured (blue dots), and fitted (Model 2, green dashed line) differential dynamic polarizability $\Delta\alpha$ between the upper ($J = 5/2$) and the lower ($J = 7/2$) clock levels; theoretical models are further described in the main text, including the Methods section. Inset: magnified view of the spectral region around the magic wavelength $\lambda_m = 813.320(6)$ nm. Error bars represent 1 s.d. (standard deviation) of the measured polarizability values

The intracavity power $P$ was determined by calibrated photo-diodes placed after the cavity outcouplers M2 and M2′. For a specified beam radius $w$ and ground state dynamic polarizability $\alpha = 195$ a.u. at $\lambda = 813$ nm[34], the lattice depth for $P = 1$ W is $U = 82E_r$, where $E_r = h \times 1785$ Hz is the recoil energy. Coefficient $\eta \leq 1$ accounts for the decrease of the average intensity for an atom with a non-zero kinetic energy. The details of the beam waist determination, power measurements, and coefficient $\eta$ calculations are given in the Methods section. Figure 2 shows the spectral dependency of $\Delta\alpha$. The magic wavelength of $\lambda_m = 813.320(6)$ nm was determined by the zero crossing of the linear fit in the inset of Fig. 2.

Trapping Tm atoms in the optical lattice at $\lambda_m$ drastically reduces inhomogeneous ac Stark broadening of the clock transition. Using excitation with a 80 ms-long Rabi π—pulses of the clock laser, we recorded a spectrum with 10 Hz full width at the half maximum shown in Fig. 3a. The nonunity excitation at the line center comes from a nonperfect initial polarization of the atoms and finite lifetime of the upper clock level ($\tau = 112$ ms).

Using a Ti:sapphire frequency comb, we determined the absolute frequency of the $|J = 7/2, F = 4\rangle \rightarrow |J = 5/2, F = 3\rangle$ clock transition in Tm of 262′954′938′269′213(30) Hz. The relative frequency uncertainty of $1.1 \times 10^{-13}$ comes mainly from the instability and calibration accuracy of a GLONASS-calibrated passive hydrogen maser used as a frequency reference for the comb.

**Differential polarizability analysis.** In the second-order approximation, the energy shift of an atomic level $|J, F, m_F\rangle$ in an external oscillating electromagnetic field with wavelength $\lambda$ equals $-\alpha_{J,F,m_F}(\lambda)E^2/4$, where $E$ is the amplitude of the electric field. For linear field polarization, the dynamic polarizability $\alpha_{J,F,m_F}$ can be split into scalar $\alpha_J^s$ and the tensor $\alpha_{J,F}^t$ parts as follows:

$$\alpha_{J,F,m_F} = \alpha_J^S + \frac{3\cos^2\Theta - 1}{2} \times \frac{3m_F^2 - F(F+1)}{F(2F-1)}\alpha_{J,F}^t, \quad (2)$$

where $\Theta$ is the angle between the quantization axis (here, the direction of the external magnetic field **B**) and the electric field polarization **ε** of the optical lattice. In our case, the differential polarizability of the two clock levels equals

$$\Delta\alpha \equiv \alpha_{5/2,3,0} - \alpha_{7/2,4,0} = \Delta\alpha^s + \frac{3\cos^2\Theta - 1}{2}\Delta\alpha^t, \quad (3)$$

where $\Delta\alpha^s = \alpha_{5/2}^s - \alpha_{7/2}^s$ and $\Delta\alpha^t = \frac{5}{7}\alpha_{7/2,4}^t - \frac{4}{5}\alpha_{5/2,3}^t$. By definition, at the magic wavelength $\lambda_m$ the differential polarizability vanishes: $\Delta\alpha(\lambda_m) = 0$.

The frequency shift of the clock transition due to the optical lattice can be caused by (i) inaccuracy of the magic wavelength determination and (ii) angular dependency of the tensor part of the differential polarizability. The accuracy of the magic wavelength determination is related to the slope of $\Delta\alpha(\lambda)$ in the vicinity of $\lambda_m$, which is $-0.075(17)$ a.u nm$^{-1}$ in Tm, as shown in the inset in Fig. 2. In dimensionless normalized units this corresponds to $\frac{\delta(\Delta\alpha)/\alpha}{h\,\delta\nu_L/E_r} \approx 1.5 \times 10^{-12}$ (here $\delta\nu_L$ is a lattice laser frequency increment), that is more than one order of magnitude smaller than the corresponding sensitivity of Sr and Yb lattice clocks[39,40].

For $\Theta \ll 1$, the $\Theta$-dependent part of the differential tensor polarizability $\Delta\alpha^t$ influences the clock transition frequency as follows:

$$h\Delta\nu \approx 3/2\Delta\alpha^t\frac{E^2}{4}\Theta^2. \quad (4)$$

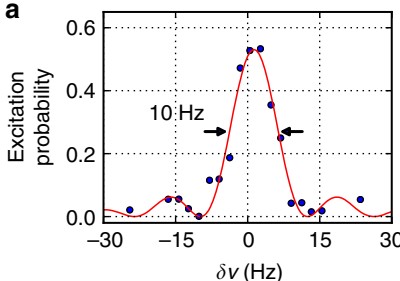

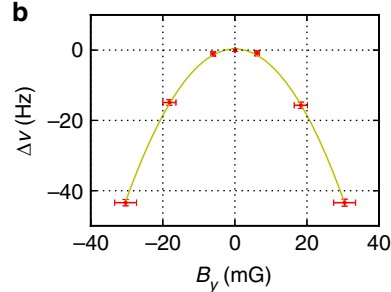

**Fig. 3** Spectroscopy of the clock transition. **a** Spectral line shape of the clock transition in Tm. Every point is an average of six measurements. The solid curve shows the fit calculated for a Fourier-limited 80 ms rectangular $\pi$-pulse. $\delta\nu$ is the detuning from an exact resonance. **b** Clock transition frequency shift $\Delta\nu$ depending on $B_y$ (dots) at $B_z = 0$ and constant $B_x = 225$ mG; the solid line is a parabolic fit. The dependence on $B_z$ is similar. Error bars represent 1 s.d. of the $B_y$ magnetic field (horizontal) and of the measured frequency shift (vertical)

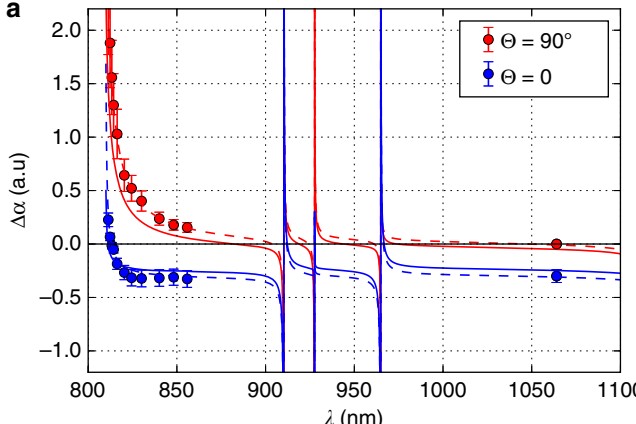

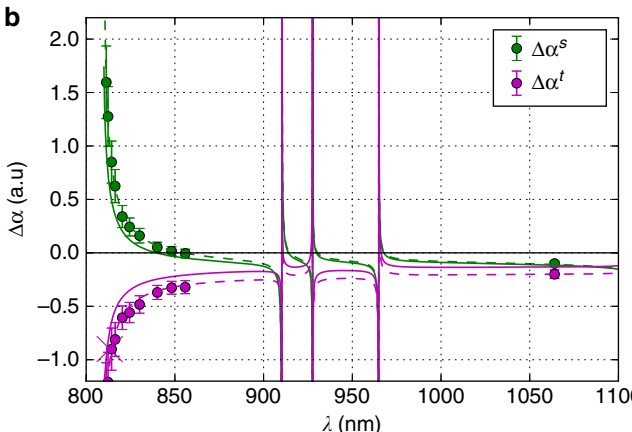

**Fig. 4** Differential polarizability spectra. **a** Differential dynamic polarizability $\Delta\alpha(\lambda)$ for $\Theta = \pi/2$ (red dots) and $\Theta = 0$ (blue dots). **b** Corresponding scalar $\Delta\alpha^s(\lambda)$ (green dots) and tensor $\Delta\alpha^t(\lambda)$ (magenta dots) parts. The magenta cross is $\Delta\alpha^t(\lambda_m)$ determined from measurements in Fig. 3b. The solid and dotted curves are calculations based on Models 1 and 2, respectively (see Methods section). Error bars represent 1 s.d. of the measured polarizability values

**Table 1 BBR shitf comparison. Fractional BBR shift at 300 K for the clock transition frequencies in thulium and selected other neutral atoms and ions**

| Element | $\Delta\nu^{BBR}/\nu$, $10^{-17}$ |
|---|---|
| Tm (this work) | 0.23 |
| Sr[a] | −550 |
| Yb[a] | −270 |
| Hg[b] | −16 |
| Yb+ [c] | −11 |
| Al+ [a] | −0.4 |
| Lu+ [d] | −0.14 |

[a]Ref. [11]
[b]Ref. [47]
[c]Transition $^2S_{1/2}$−$^2F_{7/2}$, ref. [11]
[d]Transition $^2S_{1/2}$−$^2F_{7/2}$, ref. [21]

shifts mentioned in the previous paragraph can be readily reduced to the mHz level by stabilizing the lattice wavelength with 0.1 GHz accuracy and by maintaining $|\Theta| < 10^{-3}$.

**Static differential polarizability and the BBR shift**. The BBR shift of the clock transition frequency can be accurately calculated from the static differential scalar polarizability $\Delta a^s_{DC} = \Delta a^s(\lambda \to \infty)$ using a theoretical model based on the measured polarizability spectrum at the wavelengths of 810–860 nm and at 1064 nm.

Measurements in the spectral region of 810–860 nm were done by scanning the wavelength of the Ti:sapphire laser at two polarizations corresponding to $\Theta = 0 (\varepsilon || \mathbf{x})$ and $\Theta = \pi/2 (\varepsilon || \mathbf{y})$ as shown in Fig. 4a. The corresponding scalar $\Delta\alpha^s(\lambda)$ and tensor $\Delta\alpha^t(\lambda)$ differential polarizabilities calculated from Eq. (3) are shown in Fig. 4b.

To measure the dynamic polarizability at 1064 nm, we used a slightly different procedure: Tm atoms were trapped in the optical lattice at $\lambda_m$ for which the differential polarizability vanishes, and the atomic cloud was illuminated along the $y$-axis by a focused beam of a linearly polarized single-frequency 1064 nm fiber laser with an optical power of up to 10 W. The corresponding results for $\Theta = 0 (\varepsilon || \mathbf{x})$ and $\Theta = \pi/2 (\varepsilon || \mathbf{z})$ are also shown in Fig. 4.

To compare with experimental data and to deduce $\Delta a^s_{DC}$, we use Model 2 (see Methods), which differs from Model 1 by introducing four adjustable parameters: probabilities of the 806.7 and 809.5 nm transitions and two offsets for differential scalar and tensor polarizabilities. Corresponding fits based on Model 2 are shown as dashed lines in Fig. 4, while calculations with no free parameters (Model 1) are shown as solid lines. From Model 2,

To find $\Delta\alpha^t$, we measured the dependency of the clock transition frequency shift $\Delta\nu$ for a small magnetic field $B_y$ using the constant-bias fields $B_x = 225$ mG and $B_z = 0$ ($\Theta \approx B_y/B_x$) as shown in Fig. 3b. From the corresponding parabolic coefficient of $-56(11)$ mHz mG$^{-2}$, we obtained a differential tensor polarizability of $\Delta\alpha^t = -0.9(2)$ a.u. at $\lambda_m$. The uncertainty comes from the absolute calibration of the magnetic field and power calibration (see the Methods section). Both lattice frequency

we obtain

$$\Delta \alpha_{DC}^s = -0.063(30) \text{ a.u.} \qquad (5)$$

Note, that the differential static scalar polarizability from Model 1 is $-0.062$ a.u. The BBR frequency shift can be readily calculated using the value of $\Delta a_{DC}^s$[34]. The differential scalar polarizability in the spectral region around 10 μm (the maximum of the BBR spectrum at room temperature) differs by $<10^{-3}$ a.u. from $\Delta a_{DC}^s$. Note that there are no transitions from the clock levels for $\lambda > 1.5$ μm. For the clock transition at 1.14 μm, the room temperature BBR shift is 0.60(28) mHz, which is a few orders of magnitude smaller than that for other neutral atoms and is comparable to that of the best ion species as shown in Table 1. This result quantitatively confirms the idea of strong shielding of inner-shell transitions in lanthanides from external electric fields.

## Discussion

The specific shielding of the inner-shell magnetic-dipole clock transition in atomic thulium at 1.14 μm by the outer $5s^2$ and $6s^2$ electronic shells results in a very low sensitivity of its frequency to external electric fields. The differential static scalar polarizability of the two $J = 7/2$ and $J = 5/2$ clock levels is only $-0.063(30)$ atomic units, which corresponds to the fractional BBR frequency shift of the transition of $2.3(1.1) \times 10^{-18}$ at room temperature. This amount is three orders of magnitude less than that for the prominent clock transitions in Sr and Yb (see Table 1). Unlike other systems used for optical clocks, the clock states in thulium have a large orbital momentum, which leads to tensor light shifts. Additionally, the presence of the hyperfine splitting results in the second-order Zeeman shift. Our estimations of these and other sources of the clock transition frequency shift are summarized in the Methods section and show that they can be routinely controlled to better than $10^{-17}$. Altogether, this development makes Tm a promising candidate for a transportable room-temperature optical atomic clock with $10^{-17}$ uncertainty due to soft constraints on the ambient temperature stability. This approach combines advantages of unprecedented frequency stability of optical lattice clocks on neutral atoms and low sensitivity to BBR of ion optical clocks. Moreover, precision spectroscopy in Tm opens possibilities for sensitive tests of Lorentz invariance[5] and for a search of the fine structure constant variation[41].

Optical clocks based on a $f$–$f$ transition in some other lanthanides with spinless nuclei could be even more attractive featuring the low sensitivity to magnetic fields due to the absence of the hyperfine structure and small BBR shift. For example, the fine-structure clock transition at the telecom-wavelength of 1.44 μm in laser-cooled erbium atoms[32,42] can be particularly interesting for optical frequency dissemination over fiber networks[43].

## Methods

**Enhancement cavity**. The optical lattice is formed inside a Γ-shaped enhancement cavity, as shown in Fig. 1a. The reflectivity of the curved (radius of curvature $r = -250$ mm) incoupler mirror M1 equals 87% and matches losses introduced by the vacuum chamber viewports. Outcouplers M2 or M2′ are identical flat mirrors with reflectivity $R > 99\%$. The 45° cavity mirror is mounted on a piezo actuator to lock to the laser frequency. The intracavity polarization is defined by a broadband polarization beam splitter; depending on the polarization, either the M2 or M2′ outcoupler mirror is used. The intracavity lens has a focal length of $f = 400$ mm.

Depending on the experimental geometry ($\Theta = 0$ or $\Theta = \pi/2$), we couple corresponding linearly polarized radiation from the Ti:sapphire laser through the incoupler mirror M1. Intracavity polarization filtering by PBS defines the polarization angle and significantly improves the polarization purity of the optical lattice. The angle between the laser field polarization and the bias magnetic field is adjusted with accuracy better than 1°.

**Measurement of differential dynamic polarizabilities**. The differential polarizability $\Delta \alpha$ of the clock levels is determined from the frequency shift of the corresponding transition $\Delta \nu$, circulating power $P$, and TEM$_{00}$ cavity mode radius $w$ at

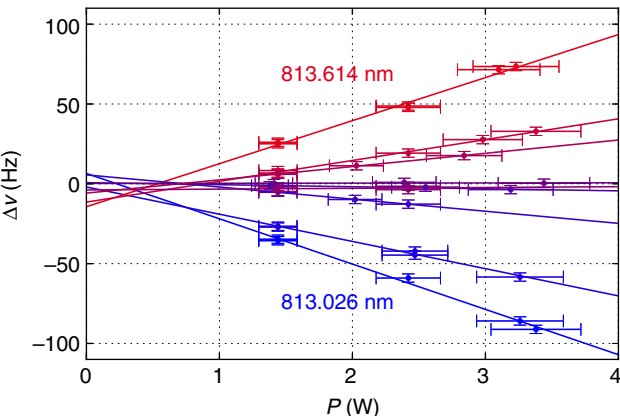

**Fig. 5** Clock transition frequency shift measurements. Clock transition frequency shift $\Delta \nu$ as a function of optical lattice power $P$ in the vicinity of the magic wavelength $\lambda_m = 813.32$ nm. The solid lines are linear fits to the experimental data. The horizontal error bars correspond to uncertainty (1 s.d.) of the power measurements, and the vertical error bars are the mean squared transition line center fit errors and 2.6 Hz (1 s.d. of the clock laser output frequency)

the atomic cloud position through Eq. (1). The dependency $\Delta \nu(P)$ was obtained for different wavelengths in the spectral range of 810–860 nm for intracavity circulating powers varying from 1 to 4 W, as shown in Fig. 5. The frequency shift $\Delta \nu$ was measured relative to the laser frequency, which is stabilized to an ultrastable ULE cavity with linear drift compensation. The slope coefficients of the corresponding linear fits were substituted into Eq. (1) to deduce $\Delta \alpha$ as presented in Figs. 2 and 4. Some residual nonlinearity and, as a consequence, imperfect intersection of linear fits at $P = 0$ can be associated with finite atomic temperature which is discussed below. Note that differential polarizabilities obtained from linear fits with an additional constraint $\Delta \nu(0) = 0$ agree with values obtained from our current analysis within one standard deviation.

Coefficient $\eta$ accounts for the effect of the reduced average light intensity $I_{av} = \eta I_0$ due to the finite temperature of the atomic cloud in the 1D optical lattice with an intensity profile $I(r, z) = I_0 e^{-2r^2/w^2} \cos^2(2\pi z/\lambda_m)$. Assuming a Boltzmann distribution of the atoms with a temperature $T$ in the trap of depth $U_0 = 100 \ldots 300 E_r$ (experimental lattice depth range), we estimate the coefficient $\eta$ using the following expression:

$$\eta = \sum_{n_z} \frac{e^{-E_{n_z}/kT}}{Z_0} \eta_z(n_z) \eta_r(n_z), \qquad (6)$$

Here the first multiplier gives the relative population of the $n_z$ axial vibrational mode with the energy $E_{n_z}$ (a number of axial vibrational states varies from 7 to 11 for $U_0 = 100 \ldots 300 E_r$), $\eta_z(n_z)$ and $\eta_r(n_z)$ are the coefficients due to delocalization in the axial and the radial directions, correspondingly. To estimate $\eta_z(n_z)$ we used numerically calculated wave functions of the axial vibrational modes in a single-well sinusoidal potential. $\eta_r(n_z)$ was calculated using classical approximation:

$$\eta_r(n_z) = \frac{\int_0^{U_0 - E_{n_z}} e^{-E/kT} \left( \frac{1}{2r_0} \int_{-r_0}^{r_0} e^{-2r^2} dr \right) dE}{\int_0^{U_0 - E_{n_z}} e^{-E/kT} dE}, \qquad (7)$$

where $r_0 = (-\ln(1 - E/U_0)/2)^{-1/2}$ is the classical turning point. We estimate parameter $\eta$ to be 0.76(15) for $kT = 0.3 U_0$, which is close to an expected atomic temperature in our experiment. The stated error includes both the uncertainty of the atomic temperature, as well as incompleteness of the model (the single-well sinusoidal potential instead of the optical lattice potential and a possible difference in radial and axial temperatures).

The uncertainty of the frequency shift $\Delta \nu$ comes from the residual instability of the reference cavity on time intervals of 1000 s. To estimate this uncertainty, we measured the clock transition frequency relative to the clock laser frequency at the magic wavelength where the perturbations from the lattice are minimal. The results are shown in Fig. 6. The standard deviation equals 2.6 Hz, contributing 0.003 a.u. to the error budget of $\Delta \alpha$. For the lattice wavelength detuned from $\lambda_m$, the contribution of the laser frequency instability is negligible.

The intracavity power $P$ was determined by measuring power leaking through the cavity outcoupler M2 (or M2′) using calibrated photodiodes. For each photodiode, we measured the power-to-voltage transfer function $P(U) = \kappa U$, where $U$ is the voltage reading from the photodiode and $\kappa$ is the coefficient measured using an absolutely calibrated Thorlabs S121C power meter. To determine $\kappa$, we unlocked the cavity, slightly tilted the outcoupler, blocked the reflected beam to

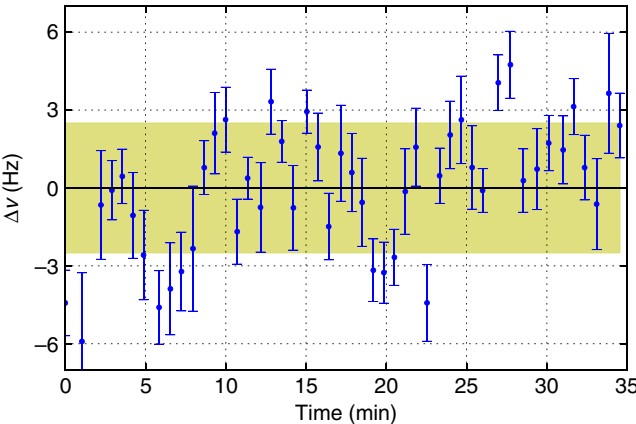

**Fig. 6** Stability of the clock laser frequency. The relative frequency of the Tm clock transition and ULE cavity mode with linear drift compensation. Each data point and corresponding uncertainty comes from the fit of the clock transition spectrum. The shaded region corresponds to 1 s.d. of the data set

prevent feasible reflections from the incoupler, and measured the power before the outcoupler and corresponding voltage reading of the photodiode. The linearity of the photodiode response was checked separately and was found to be better than 3% in the working region. The photodiode calibration was done in the whole spectral range of 810–860 nm to take into account the spectral response of the outcoupler and the photodiode. Although the specified uncertainty of the power sensor Thorlabs S121C equals 3%, we estimate the net uncertainty of power measurement to be 10% based on a comparison of readings from three different absolutely calibrated sensors.

The beam radius $w$ at the atomic cloud position is deduced from the cavity geometry: distances from the vacuum chamber center (and target atomic cloud position) to M1, L and M2 (or M2′) are 244, 384, and 500 mm, respectively, giving a beam radius of $w = 126$ μm at the position of atomic cloud. The uncertainty of $w$ comes from the position uncertainties of cavity elements and of the atomic cloud with respect to the chamber center, in addition to the uncertainty of 1 mm of the lens focal length. We conservatively evaluate the position uncertainties of M1, L, and M2 (or PBS and M2′) as 1 mm and the possible axial displacement of the atomic cloud of 2 mm. The partial contributions to the beam radius uncertainty are 1.6 μm (the incoupler), 1.2 μm (the lens), 0.03 μm (the outcoupler), 0.3 μm (the cloud), and 1.5 μm (focal length). Adding up in quadratures, the total uncertainty of the beam radius $w_0$ equals 2.5 μm. The result is independently confirmed by measuring frequency intervals between cavity transversal modes.

Summarizing, we evaluate the uncertainty of measured differential polarizability $\Delta\alpha$ to be 22% with the dominating contribution from the lattice intensity uncertainty. In the vicinity of $\lambda_m$, the uncertainty is slightly higher by 0.003 a.u. because of the reference clock laser frequency instability.

To measure differential polarizability at 1064 nm, atoms were trapped in the optical lattice at the magic wavelength and irradiated by focusing a slightly elliptic 1064 nm laser beam with waists of $w_x = 320(20)$ μm and $w_y = 280(20)$ μm in the $x$ and $y$ directions, respectively. To adjust the 1064 nm laser beam center to the atomic cloud, we maximized the intensity of the beam on the atomic cloud by monitoring the frequency shift of the clock transition. To increase the sensitivity, corresponding measurements were performed by tuning the clock laser to the slope of the 1.14 μm transition.

For the measurement session of $\Delta\alpha$ at $\Theta = 0$, we performed three adjustments of the 1064 nm laser beam. The reproducibility corresponded to a frequency shift of 5 Hz at the maximum frequency shift of 25 Hz. The resulting linear coefficient was evaluated as $\Delta\nu/P = 3.1(6)$ Hz W$^{-1}$ including this uncertainty. In the $\Theta = \pi/2$ configuration, we did not observe any significant effect from adjustment; the coefficient was $-0.04(24)$ Hz W$^{-1}$.

**Theoretical analysis**. The theoretical approach to calculate the polarizabilities of Tm atomic levels was described in our previous works[34,44]. These calculations are based on the time-dependent second-order perturbation theory with summation over known discrete transitions from the levels of interest. For calculations, we used transitions wavelengths and probabilities obtained with the COWAN package[38] with some exceptions: for transitions with $\lambda > 800$ nm, experimental wavelengths from ref. [45] are used. This approach allows the accuracy of the magic wavelength prediction to be increased in the corresponding spectral region of $\lambda > 800$ nm. According to calculations, the magic wavelength was expected at 811.2 nm, which motivated our experimental studies in the spectral region 810–815 nm (see Fig. 2). We refer to this model as Model 1 and use it for comparison with the experimental results of this work as shown in Figs. 2 and 4.

The deviation of the experimental data from Model 1 can be explained by two main factors. First, we did not take into account transitions to the continuum in this model. Together with uncertainties of COWAN calculations of transition amplitudes, this aspect can result in a small offset of the infrared differential polarizability spectrum. Note that although transitions to the continuum spectrum may contribute to polarizabilities of the individual levels (up to 10%), for the differential polarizability of the $f$–$f$ transition at 1.14 μm in the Tm contribution of the continuum is mostly canceled[34].

To fit experimental data, we use Model 2, which differs from Model 1 through the introduction of four fitting parameters. For these parameters, we use the probabilities of the 806.7 and 809.5 nm transitions, which mostly affect the polarizability spectrum in the 810–860 nm region and the two offsets for the scalar and tensor polarizabilities. After fitting the experimental data (see Fig. 4) by Model 2, the probability of the 806.7 nm transition is changed from 3473 to 4985(1100) s$^{-1}$, the probability of the 809.5 nm transition is changed from 149 to 424(70) s$^{-1}$, and the fitted offsets for the differential scalar and tensor polarizabilities equal $-0.006$ (30) and $-0.074(30)$ a.u.

The transitions from the upper Tm clock level $J = 5/2$ in the spectral range $\lambda > 900$ nm are weak, and their probabilities are not experimentally measured. To calculate probabilities, we used the COWAN package. To estimate the impact of insufficient knowledge of the transition probabilities on the differential scalar polarizability $\Delta\alpha^s(\lambda)$, we assume a potential variation of each transition probability by a factor of 2. After extrapolation of the fitted Model 2 to $\lambda \to \infty$, we obtain the static differential polarizability $\Delta\alpha^s_{DC} = -0.063^{+0.01}_{-0.005}$ a.u., where the uncertainty comes from the variation of the $\lambda > 900$ nm transition probabilities.

We summarize all sources of uncertainty that contribute to the error of the static differential polarizability $\Delta a^s_{DC}$ in Table 2. As discussed above, the experimental uncertainty for the 810–860 nm range contributes 0.025 a.u. to the $\Delta a_{DC}$, while the error in the measurement at 1064 nm results in 0.014 a.u. variation of Model 2 extrapolation. The uncertainty of the angle $\Theta$ adjustment contributes 0.002 a.u. The uncertainty associated with the poorly known transition probabilities from the $J = 5/2$ clock level in $\lambda > 900$ nm range contributes 0.01 a.u. Using extrapolation of Model 2 and adding all uncertainties, we obtain the final result of $\Delta a^s_{DC} = -0.063(30)$ a.u. Note that this finding is fully consistent with the extrapolated value of $-0.062$ a.u. obtained from Model 1 and given for the reference in Table 2.

**Frequency uncertainty estimation**. Here, we discuss expected sources of the 1.14 μm clock transition frequency shift and uncertainty. We evaluate necessary control over the experimental parameters to maintain corresponding uncertainty below 1 mHz, or $4 \times 10^{-18}$ in fractional units.

Zeeman shift: The frequency of the $|F = 4, m_F = 0\rangle \to |F = 3, m_F = 0\rangle$ clock transition possesses a quadratic sensitivity to a dc magnetic field $B$ with a coefficient $\beta = 257.2$ Hz G$^{-2}$[34]. To provide uncertainty of the transition frequency below 1 mHz, it would be necessary to stabilize the magnetic field at the level of 20 μG at the bias field of $B = 100$ mG. However, the quadratic Zeeman shift in Tm can be fully canceled by measuring an averaged frequency of two clock transitions $|F = 4, m_F = 0\rangle \to |F = 3, m_F = 0\rangle$ and $|F = 3, m_F = 0\rangle \to |F = 2, m_F = 0\rangle$ (Fig. 1b) with the quadratic Zeeman coefficients of the opposite signs. The magic wavelength condition is simultaneously fulfilled for both transitions since $\alpha_{JJ-1/2,0} \equiv \alpha_{JJ+1/2,0}$ for atoms with $I = 1/2$ (index reads as $|J, F, m_F\rangle$).

Lattice shift: The clock transition frequency shift due to the optical lattice was evaluated in the Differential polarizability analysis section. While the slope of the differential polarizability at the magic wavelength is less than that of the optical clock using other neutral atoms, the differential tensor polarizability of 0.9(2) a.u. imposes strict requirements on the quantization axis alignment and the lattice light polarization purity. Control of $|\Theta| < 10^{-3}$ (which provides $\Delta\nu < 1$ mHz according to Eq. (6) for a lattice depth of $100E_r$) can be achieved through a calibration procedure similar to the one described in Fig. 3b. The magnetic field should be stabilized at the level $10^{-3}B$, or 0.1 mG for the bias field of 100 mG.

Higher-order lattice shifts: Our estimations (using the method described in ref. [34]) show that the hyperpolarizability shift at $100Er$ lattice depth should be $<10^{-17}$ in fractional units. Moreover, both hyperpolarizability and M1 and E2

**Table 2 Uncertainty budget for the differential scalar polarizability. All sources of uncertainty that contribute to the error of the static differential polarizability $\Delta a^s_{DC}$**

| Source | Uncertainty (a.u.) |
| --- | --- |
| Experimental results for 810–850 nm | 0.025 |
| Experimental result for 1064 nm | 0.0014 |
| Angle $\Theta$ | 0.002 |
| Transition probabilities for $\lambda > 900$ nm | 0.01 |
| *Total* | 0.030 |
| For reference: | |
| Difference between Models 1 and 2 | 0.001 |

transition contribution shifts can be suppressed by reducing the trap depth or by using the operational wavelength[40].

Since the electric field gradient in the optical lattice oscillates at the optical frequency, the time-averaged frequency shift associated with a permanent differential quadrupole moment of the clock transition is zero.

BBR electric field: The 1.14 μm clock transition frequency shift due to the electric component of BBR at room temperature is 0.60(28) mHz, with uncertainty coming from the polarizability measurement error. Accordingly, even approximate stabilization of the ambient temperature would provide a clock performance of much better than $10^{-17}$.

BBR magnetic field: To estimate the clock transition frequency shift due to the magnetic component of BBR, we follow the analysis given in the work[46]. The corresponding frequency shift of one of the clock levels coupled to another atomic level with magnetic-dipole transition at frequency $\omega_0$ can be found by integrating over the full BBR spectrum as follows:

$$\Delta\nu_{\mathrm{bbr}}^{\mathrm{B}}(T) = -\frac{\omega_0}{2\pi}\frac{\mu_{\mathrm{B}}^2}{2\hbar\pi^2 c^5 \varepsilon_0}\int_0^\infty \frac{1}{\omega_0^2-\omega^2}\frac{\omega^3}{e^{\hbar\omega/kT}-1}\mathrm{d}\omega$$
$$= -\frac{\omega_0}{2\pi}\frac{\gamma}{2}\left(\frac{T}{T_0}\right)^2 f(y), \tag{8}$$

where $\varepsilon_0$ is the vacuum permittivity, $T_0 = 300$ K, and $y = \hbar\omega_0/k_{\mathrm{B}}T$. Here,

$$\gamma = \frac{\mu_{\mathrm{B}}^2}{\hbar^2}\frac{\hbar}{6c^5\varepsilon_0}\left(\frac{k_{\mathrm{B}}T_0}{\hbar}\right)^2 \approx 9.78\times 10^{-18}, \tag{9}$$

$$f(y) = \frac{6}{\pi^2}\int_0^\infty \frac{1}{y^2-x^2}\frac{x^3\mathrm{d}x}{e^x-1}. \tag{10}$$

The hyperfine transition frequency $\omega_0$ of the $J = 7/2$ ground level in Tm equals $2\pi \times 1496$ MHz, while for the $J = 5/2$ clock level it equals $2\pi \times 2115$ MHz. For these values of $\omega_0$, $y \ll 1$, $f(y) \approx -1$, and the shift is on the order of $10^{-8}$ Hz. To estimate the contribution from the optical transitions, we evaluate the shift from the lowest frequency magnetic-dipole transition at $\omega_0 \approx 2\pi \times 263$ THz, which is the clock transition itself: $y = 42$, $f(y) = 2.3\times10^{-3}$, and $\Delta\nu_{\mathrm{bbr}}^{\mathrm{B}}(T_0) = -3\times10^{-6}$ Hz for the ground level (for the upper clock level, the corresponding shift is $+3\times10^{-6}$ Hz). Hence, we estimate the total shift of the clock transition from the magnetic component of BBR to be $<10^{-4}$ Hz, or $<10^{-18}$ in fractional units.

Stray electric fields: The differential static polarizability of the clock transition does not exceed 0.2 a.u. for any angle $\Theta$, which means that static stray fields of 1 kV m$^{-1}$ would produce a fractional frequency shift smaller than $1\times10^{-17}$. In turn, there is a differential quadrupole moment of clock levels (with an upper limit of 1 a. u.[34]), which may lead to the static quadrupole frequency shift. Accordingly, one needs a stray electric field gradient of 5 kV m$^{-2}$ to reach a fractional frequency shift of $1\times10^{-17}$. In regular lattice experiments, both stray electric fields and their gradients are much smaller than the abovementioned values.

## Data availability

All data presented and analyzed in this study are available from the corresponding author upon reasonable request.

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

## Acknowledgements

Development of the ultrastable clock laser is financially supported by RFBR grant no. 16-29-11723. V.S. acknowledges financial support from RFBR grant no. 18-02-00628. A.G., E.F., D.T., K.K., and N.K. acknowledge support from RSF grant no. 19-12-00137. We are grateful to S. Kanorski and V. Belyaev for invaluable technical support and to S. Fedorov for assembling the frequency comb. We also thank M. Barrett for very useful discussions.

## Author contributions

A.G., E.F. and D.T. carried out all measurements and analyzed the data; A.G. and D.S. performed the theoretical calculations; and A.G., D.S. and N.K. wrote the paper. K.K., V.S. and N.K. conceived and directed the project. All authors discussed the results and commented on the manuscript.

## Additional information

**Competing interests:** The authors declare no competing interests.

**Journal Peer review Information:** *Nature Communications* thanks the anonymous reviewers for their contribution to the peer review of this work. Peer reviewer reports are available.

