## [Peer Review File · Nature Communications]

Reviewers' comments:

Reviewer #1 (Remarks to the Author):

The authors demonstrate the thulium optical lattice clock by carefully determining the magic wavelength, tensor shift sensitivity, quadratic Zeeman shift, and the BBR sensitivity. The authors claim very small BBR shift makes Tm optical clock one of the best candidates for transportable optical clocks with 10^{-17} uncertainty.

To the best of my knowledge, this is the first demonstration of the optical lattice clock on the f-f transition that is expected to reduce uncertainties arising from BBR and collisional shifts. Experimental procedures, measurements, and results are well written. Important parameters, such as the magic wavelength, differential polarizabilities, and BBR shift are carefully measured and presented.

The major claims of Tm clock with very small BBR shift is novel and convincingly demonstrated in the paper, which may influence new thinking in the clock community. In the framework of authors' goal of achieving 10^{-17} accurate clock, the paper is indeed self-sufficient.

Although the authors tightly focus on "orders of magnitude smaller BBR shift" in Tm, eventual performance of atomic clocks is determined by the sum of all potential uncertainties. Compared with the other optical lattice clocks operating on J=0-J=0 transitions, Tm clock shows orders of magnitude larger tensor shift and second order Zeeman shift, which will finally limit the performance of Tm clock. Authors should carefully address the issue that small BBR shift alone does not guarantee a good clock and that there are trade-offs. Otherwise general readers will be misled. In recent literatures the BBR shifts are highlighted, this is because the other systematic uncertainties, such as lattice light shifts, collisional shifts, and Zeeman shifts are well controlled and suppressed.

From the novelty point of view, the manuscript is acceptable after some revisions as mentioned above and as follows.

1)Page 3 right middle: To reduce the 2nd order Zeeman effect, the authors propose the magic wavelength where tensor shift is tuned to zero to interrogate 2 clock transitions. At this magic wavelength, how good control of θ , which corresponds to control of ambient magnetic field, is required? The angle sensitivity would be much larger than that given in Eq. (3).

2)Page 6, left middle: With η parameter ~ 0.9 given in (6), how good can one control the hyperpolarizability effect? Is it still in the 10^{-17} target?

3)Page 4, right: To support the discussion for the uncertainties, including above issues, uncertainty budgets will be helpful.

4)Are the M1, E2 dynamic polarizabilities small enough to achieve 10^{-17} uncertainty?

Reviewer #2

Referee report on manuscript NCOMMS-18-31452

This paper reports investigations related to a new concept for an optical clock based on atoms prepared in an optical lattice. The novelty lies in the use of thulium (Tm) atoms, which offer some qualitative differences and advantages over other atomic systems that have been considered so far for this purpose. The investigations mainly deal with the determination of the magic wavelength of the lattice that eliminates the dynamic Stark shift of the clock transition frequency, and with the determination of the Stark shift due to the ambient blackbody radiation. This shift is found to be much smaller than in other neutral atoms that are presently investigated for optical clocks.

I recommend publication in Nature Communications because the results presented in this paper have a significant novelty value. Most details of the presentation appear to be clear and adequate. Nevertheless there are some points where additional information should be given or where the text should be changed. These items are noted in the following.

Major points of criticism:

(1) Page 1, second paragraph: A "fractional frequency instability" at the low 10^{-18} level was demonstrated for Sr and Yb optical lattice clocks, but *not* (at least not directly) for single-ion clocks since this would require very long averaging times. What was demonstrated for all the mentioned clocks is a *systematic uncertainty* ("accuracy") in this range. Probably this is also the intended statement, because the discussion in this paragraph concentrates on effects that potentially limit the accuracy.

(2) Page 4, discussion around Eq. (4) and page 7, "electric quadrupole shift": Information on the magnitude of the *tensorial* part of the differential static polarizability is missing. This should be given in order to estimate the sensitivity of the clock transition to static electric stray fields. (For Sr and Yb lattice clocks, the stray field originating from glass windows can be a serious perturbation.) In the case of Tm, also the quadrupole shift caused by an inhomogeneous stray field would need to be considered.

(3) Page 5, Figure 5 and related discussion: It is obvious from Fig. (5) that the extrapolations of the linear fits of clock transition frequency $\Delta\nu$ as a function of lattice power P fail to come close to the point ($P = 0, \Delta\nu = 0$) for those wavelengths that have a significant offset from the magic wavelength. Instead, the extrapolations intersect in the range $P = 0.5 \dots 1$ W, which is not expected if the light shift is proportional to the applied lattice light power. Probably this feature does not call into question the

inferred value of the magic wavelength, but it seems to be ignored in the uncertainty analysis that determines the size of the error bars of the off-center data points in the inset of Fig. 2. It seems possible (among other explanations) that this feature is related to a temperature-dependent variation of the average light intensity seen by the atoms when the lattice potential depth is varied. The authors model this effect in Eq. (6). However, the data of Fig. 5 are not discussed in the context of Eq. (6) and no effort is made to apply corrections. In any case, the authors should try to explain the extrapolation intersection defect in Fig. 5 and take it into account in the uncertainty analysis related to the magic wavelength.

Minor points of criticism:

(4) Even if there is only one stable Tm isotope, its mass number (169) and nuclear spin should be noted in the beginning.

(5) Throughout the text, there are a number of minor language imperfections, mostly related to definite/indefinite articles and plural/singular form (pulse/pulses). Careful independent or editorial proofreading is recommended.

(6) Page 3, paragraph at top of right column: The expression for the clock transition frequency shift, " $U \times \Delta f \times 0.30(4)$ mHz for.... Δf [GHz] ..." is difficult to read and might be formally inconsistent. A better way to write this would be, e.g., "transition frequency shift coefficient $\Delta\nu/\Delta f = 0.30(4) \times 10^{-XX} U$ "

(7) Page 6, paragraph at bottom of right column: The "probability of the 806.7 nm transition" is noted with the physical unit of a rate, s^{-1} . Probably the numbers mentioned are the spontaneous decay rates of the upper level of the mentioned transitions, possibly (or not) containing a branching ratio factor. Instead of this unclear term, better refer to spontaneous decay rates or give the values of the corresponding oscillator strengths.

Reviewer #3

In their manuscript, Golovizin and coworkers examine the narrow $1.14 \mu\text{m}$ optical transition in neutral thulium. Ultimately, the goal is to realize a high-performance optical lattice clock based on this transition. To the best of my knowledge, all other high-performance neutral atom optical clocks, demonstrated or currently being pursued, are based on the ground 1S_0 to metastable 3P_0 transition in alkaline earth or alkaline earth-like atoms (e.g., Sr and Yb). The clock transition considered here stands in stark contrast, which has both benefits and drawbacks. The most notable benefit is the inherent suppression of the blackbody radiation (BBR) shift, found here to be a few orders of magnitude smaller than the conventional clock transitions.

This same group (at least a large overlap in authors) considered this clock transition previously in reference [34]. While that work contained experimental aspects, it was primarily a theoretical study. This work is primarily experimental, making strides towards the goal stated above. The BBR shift, the magic wavelength, and the clock frequency are evaluated to respectable precision for a pioneering work. As best as I can tell, the experimental methods and stated uncertainties seem reasonable.

As mentioned, the chosen transition has some drawbacks compared to the conventional $^1S_0 \rightarrow ^3P_0$ clock transitions. Mostly this stems from the $J \neq 0$ character of the clock states and the natural lifetime of the upper clock state. I am skeptical that the proposed clock will ever reach the performance levels already demonstrated by Sr and Yb optical lattice clocks (low- 10^{-18} accuracy, sub- 10^{-18} stability), but the authors do not make such a claim. All claims regarding the prospects of the clock seem well-grounded and appropriate.

I believe this work is well-motivated. As correctly noted in the manuscript, the BBR shift is a major concern for state-of-the-art optical clocks, especially the optical lattice clocks based on neutral atoms. I find the three order-of-magnitude suppression for a neutral system, as demonstrated here, to be rather astounding. As the authors point out, this could prove useful for, say, a transportable clock. One might speculate that other similar (but narrower) transitions could be identified in other neutral atoms and perhaps “tricks” could be developed to circumvent, e.g., lattice polarization sensitivity (one of the drawbacks associated with $J \neq 0$). In any case, I’m intrigued. Given consideration of the comments below, I recommend publication in Nature Communications.

- For the most part, the manuscript is well-organized and clear. My biggest complaint is grammatical errors/nuances that appear throughout (I suspect the authors are not native English speakers, which deserves some forgiveness in this regard). In the Abstract alone, I point to phrases “BBR shift at the room temperature,” “smaller compared to,” “optical clocks on neutral atoms,” and “uncertainty in 10^{-17} level.”
- One page 2 (also in Methods, Eq. (5)), where the equation $h\Delta\nu = -16a_0^3\Delta\alpha P/cw^2$ appears, it should be clearly stated that $\Delta\alpha$ in *this* equation is the differential polarizability *in atomic units*. That is, here $\Delta\alpha$ is a number that corresponds to the ratio of the differential polarizability to the atomic unit for polarizability (in one form, $4\pi\epsilon_0 a_0^3$). Elsewhere in the manuscript, the symbol $\Delta\alpha$ is just the differential polarizability. For example, a couple of sentences later “ $\Delta\alpha$ in atomic units” is intended to be read “the differential polarizability in atomic units.” In my opinion, the most palatable option would be to rewrite the equation as $h\Delta\nu = -16\Delta\alpha P/4\pi\epsilon_0 cw^2$, with $\Delta\alpha$ consistently representing the differential polarizability throughout (and it can be reported in whatever units you want; Figure 2, for example, would be perfectly acceptable).
- I don’t recall seeing trap depths ever given in recoil energy. This is fairly standard for this field, as it is the natural unit for motional effects in the lattice (at least in the longitudinal direction).
- I did not attempt to verify equation (6) of the Methods. Still, I am surprised not to see some account of the density of motional states in the lattice (which I think would at least be apparent in the denominator). Note that the density of states is greater near the top of the trap ($E \rightarrow U_0$) than near the bottom ($E \rightarrow 0$). In line with the previous comment, knowing the depth in recoil energy would help a reader gauge whether or not a classical treatment of the motion in the trap is reasonable.
- The colloquial abbreviation “BBR” should be written out as “blackbody radiation” in the title.

Dear Reviewers,

we are thankful for your recommendation to publish our manuscript in Nature Communications, as well as for valuable comments and suggestions. We have revised our manuscript accordingly.

Please find our response to the comments below.

Response to Reviewer #1

The authors demonstrate the thulium optical lattice clock by carefully determining the magic wavelength, tensor shift sensitivity, quadratic Zeeman shift, and the BBR sensitivity. The authors claim very small BBR shift makes Tm optical clock one of the best candidates for transportable optical clocks with 10^{-17} uncertainty.

To the best of my knowledge, this is the first demonstration of the optical lattice clock on the f-f transition that is expected to reduce uncertainties arising from BBR and collisional shifts. Experimental procedures, measurements, and results are well written. Important parameters, such as the magic wavelength, differential polarizabilities, and BBR shift are carefully measured and presented.

The major claims of Tm clock with very small BBR shift is novel and convincingly demonstrated in the paper, which may influence new thinking in the clock community. In the framework of authors goal of achieving 10^{-17} accurate clock, the paper is indeed self-sufficient.

Although the authors tightly focus on orders of magnitude smaller BBR in Tm, eventual performance of atomic clocks is determined by the sum of all potential uncertainties. Compared with the other optical lattice clocks operating on J=0-J=0 transitions, Tm clock shows orders of magnitude larger tensor shift and second order Zeeman shift, which will finally limit the performance of Tm clock. Authors should carefully address the issue that small BBR shift alone does not guarantee a good clock and that there are trade-offs. Otherwise general readers will be misled. In recent literatures the BBR shifts are highlighted, this is because the other systematic uncertainties, such as lattice light shifts, collisional shifts, and Zeeman shifts are well controlled and suppressed.

We agree with the referee that Tm clocks might suffer from other sources of uncertainties, like tensor light shift and second order Zeeman shift. In the manuscript, it was shown that control over these effects at the 1 mHz level can be easily achieved (the last two paragraphs before Section "Static differential polarizability and the BBR shift"). We have updated the manuscript to briefly discuss major error sources in Methods and show that every one of them can be controlled at least at 1 mHz level. This makes realization of portable clocks with 10^{-17} uncertainty feasible. We also altered Discussion section to clarify this:

The specific shielding of the inner-shell magnetic-dipole clock transition in atomic thulium at 1.14 μm by the outer $5s^2$ and $6s^2$ electronic shells results in a very low sensitivity of its frequency to external electric fields. The differential static scalar polarizability of the two $J = 7/2$ and $J = 5/2$ clock levels is only $-0.047(18)$ atomic units, which corresponds to the fractional BBR frequency shift of the transition of $1.7(7) \times 10^{-18}$ at room temperature. This amount is three orders of magnitude less than that of the

prominent clock transitions in Sr and Yb (see Table 1). Unlike other systems used for optical clocks, the clock states in thulium have a large orbital momentum, which leads to tensor light shifts. Additionally, the presence of the hyperfine splitting results in the second-order Zeeman shift. Our estimations of these and other sources of the clock transition frequency shift are summarized in the Methods section and show that they can be routinely controlled to better than 10^{-17} . Altogether, this development makes Tm a promising candidate for a transportable room-temperature optical atomic clock with 10^{-17} uncertainty due to soft constraints on the ambient temperature stability.

From the novelty point of view, the manuscript is acceptable after some revisions as mentioned above and as follows.

1) Page 3 right middle: To reduce the 2nd order Zeeman effect, the authors propose the magic wavelength where tensor shift is tuned to zero to interrogate 2 clock transitions. At this magic wavelength, how good control of Θ , which corresponds to control of ambient magnetic field, is required? The angle sensitivity would be much larger than that given in Eq. (3).

The referee is correct that control over non-zero angle Θ is more challenging. After submitting the manuscript, we found that for $m_F = 0$ sublevels of the hyperfine levels in atoms with $I = 1/2$ the polarizabilities are identical. This results in fulfilling the magic wavelength condition for both mentioned transitions at $\lambda_m \approx 813.3$ nm for $\Theta = 0$, thus angle dependence given in Eq. 3 remains valid.

We changed the manuscript accordingly and moved the corresponding paragraph to Section "Frequency uncertainty estimation" in Methods :

To provide uncertainty of the transition frequency below 1 mHz, it would be necessary to stabilize the magnetic field at the level of $20 \mu\text{G}$ at the bias field of $B = 100 \text{ mG}$. However, the quadratic Zeeman shift in Tm can be fully canceled by measuring an averaged frequency of two clock transitions $|\mathbf{F} = 4, \mathbf{m}_F = 0\rangle \rightarrow |\mathbf{F} = 3, \mathbf{m}_F = 0\rangle$ and $|\mathbf{F} = 3, \mathbf{m}_F = 0\rangle \rightarrow |\mathbf{F} = 2, \mathbf{m}_F = 0\rangle$ [Fig. 1(b)] with the quadratic Zeeman coefficients of the opposite signs. The magic wavelength condition is simultaneously fulfilled for both transitions since $\alpha_{J, J-1/2, 0} \equiv \alpha_{J, J+1/2, 0}$ for atoms with $I = 1/2$ (index reads as $|\mathbf{J}, \mathbf{F}, \mathbf{m}_F\rangle$).

2) Page 6, left middle: With η parameter 0.9 given in (6), how good can one control the hyperpolarizability effect? Is it still in the 10^{-17} target?

4) Are the M1, E2 dynamic polarizabilities small enough to achieve 10^{-17} uncertainty?

We thank the reviewer for pointing out these possible sources of uncertainty. We added comments on these sources of the clock transition frequency shift to Section "Frequency uncertainty estimation" in Methods (see below). We do not expect strong resonant contribution from M1 and E2 transitions at 813 nm, and the estimations show that the non-resonant part should contribute to the clock transition frequency shift at the level less than 10^{-17} . A more careful investigation of these effects should be a topic of a separate study.

Higher-order lattice shifts. Our estimations (using the method described in [34]) show that the hyperpolarizability shift at $100 E_r$ lattice depth should be less than 10^{-17} in fractional units. Moreover, both hyperpolarizability and M1 and E2 transition contribution shifts can be suppressed by reducing the trap depth or by using the operational wavelength [40].

3) Page 4, right: To support the discussion for the uncertainties, including above issues, uncertainty budgets will be helpful.

We are grateful to the reviewer for this comment, the authors discussed exactly this issue during manuscript preparation as well. In our opinion, the table of uncertainties may give to a reader a false impression that we already experimentally characterized Tm-based optical clocks. We decided not to declare the full uncertainty budget in the main text and in the form of table as it requested in typical metrological papers. Still, to support our proposition of achievable 10^{-17} uncertainty we discuss all major sources in "Frequency uncertainty estimation" Section in Methods.

Response to Reviewer #2

This paper reports investigations related to a new concept for an optical clock based on atoms prepared in an optical lattice. The novelty lies in the use of thulium (Tm) atoms, which offer some qualitative differences and advantages over other atomic systems that have been considered so far for this purpose. The investigations mainly deal with the determination of the magic wavelength of the lattice that eliminates the dynamic Stark shift of the clock transition frequency, and with the determination of the Stark shift due to the ambient blackbody radiation. This shift is found to be much smaller than in other neutral atoms that are presently investigated for optical clocks.

I recommend publication in Nature Communications because the results presented in this paper have a significant novelty value. Most details of the presentation appear to be clear and adequate. Nevertheless there are some points where additional information should be given or where the text should be changed. These items are noted in the following.

Major points of criticism:

1) Page 1, second paragraph: A "fractional frequency instability" at the low 10^{-18} level was demonstrated for Sr and Yb optical lattice clocks, but not (at least not directly) for single-ion clocks since this would require very long averaging times. What was demonstrated for all the mentioned clocks is a systematic uncertainty ("accuracy") in this range. Probably this is also the intended statement, because the discussion in this paragraph concentrates on effects that potentially limit the accuracy."

We thank reviewer for this comment which is absolutely correct for the cited works in our manuscript. The rewritten sentence is:

*The fractional frequency **uncertainty** of optical clocks reaching a low level of 10^{-18} was successfully demonstrated for Sr ...*

2) Page 4, discussion around Eq. (4) and page 7, "electric quadrupole shift": Information on the magnitude of the tensorial part of the differential static polarizability is missing. This should be given in order to estimate the sensitivity of the clock transition to static electric stray fields. (For Sr and Yb lattice clocks, the stray field originating from glass windows can be a serious perturbation.) In the case of Tm, also the quadrupole shift caused by an inhomogeneous stray field would need to be considered.

We thank the reviewer for pointing out these possible sources of the clock transition frequency shift. We added the following sentences to the Methods section:

The differential static polarizability of the clock transition does not exceed 0.2 a.u. for any angle Θ , which means that static stray fields of 1 kV m^{-1} would produce a fractional frequency shift smaller than 1×10^{-17} . In turn, there is a differential quadrupole moment of clock levels (with an upper limit of 1 a.u. [34]) which may lead to the static quadrupole frequency shift. Accordingly, one needs a stray electric field gradient of 5 kV m^{-2} to reach fractional a frequency shift of 1×10^{-17} . In regular lattice experiments, both stray electric fields and their gradients are much smaller than the abovementioned values.

3) Page 5, Figure 5 and related discussion: It is obvious from Fig. (5) that the extrapolations of the linear fits of clock transition frequency as a function of lattice power P fail to come close to the point ($P = 0, \Delta\nu = 0$) for those wavelengths that have a significant offset from the magic wavelength. Instead, the extrapolations intersect in the range $P = 0.5 \dots 1 \text{ W}$, which is not expected if the light shift is proportional to the applied lattice light power. Probably this feature does not call into question the inferred value of the magic wavelength, but it seems to be ignored in the uncertainty analysis that determines the size of the error bars of the off-center data points in the inset of Fig. 2. It seems possible (among other explanations) that this feature is related to a temperature-dependent variation of the average light intensity seen by the atoms when the lattice potential depth is varied. The authors model this effect in Eq. (6). However, the data of Fig. 5 are not discussed in the context of Eq. (6) and no effort is made to apply corrections. In any case, the authors should try to explain the extrapolation intersection defect in Fig. 5 and take it into account in the uncertainty analysis related to the magic wavelength.

We are very grateful to the reviewer for this comment. Indeed, the η coefficient ($I_{av} = \eta I_0$) can be different for different lattice depths, so one should take this fact into account when slopes $\Delta\nu(P)$ are determined. In first two figures below (a, b), we added the uncertainty of η together with 2.6 Hz standard deviation

of $\Delta\nu$ measurement which is discussed in Methods (Fig. 6). The third figure (c) compares results of two fit procedures: regular linear regression of each data set (panel a) and with fixed value $\Delta\nu(P=0) = 0$ (panel b). One can see that slope coefficients from these fits coincide within corresponding standard deviation.

Figure 1: Updated plots of the clock transition frequency shift measurement with free (a) and fixed (b) values of $\Delta\nu(P=0) = 0$ in the linear fits. (c) Plot of differential polarizabilities deduced from (a, blue) and (b, red).

We replaced Fig. 5 in the Methods by the left (a) figure from above, the paragraph after eq. 5 is also changed accordingly:

*The dependency $\Delta\nu(P)$ was obtained for different wavelengths in the spectral range of 810–860 nm for intracavity circulating powers varying from 1 W to 4 W as shown in Fig. 5. The frequency shifts $\Delta\nu$ were measured relative to the laser frequency, which is stabilized to an ultrastable ULE cavity with linear drift compensation. The slope coefficients of the corresponding linear fits were substituted into Eq. (5) to deduce $\Delta\alpha$ as presented in Figs. 2 and 4. **Some residual nonlinearity and, as a consequence, imperfect intersection of linear fits at $P = 0$ can be associated with finite atomic temperature. The influence of the atomic temperature was analyzed further (Eq. 6). Note that differential polarizabilities obtained from linear fits with an additional constraint $\Delta\nu(0) = 0$ agree with values obtained from our current analysis within one standard deviation.***

Minor points of criticism:

4) Even if there is only one stable Tm isotope, its mass number (169) and nuclear spin should be noted in the beginning.

We fully agree with the reviewer and mentioned this fact it in the manuscript.

5) Throughout the text, there are a number of minor language imperfections, mostly related to definite/indefinite articles and plural/singular form (pulse/pulses). Careful independent or editorial proof-reading is recommended.

We asked a native speaker to double-check English over the manuscript.

6) Page 3, paragraph at top of right column: The expression for the clock transition frequency shift, " $U \times \Delta f \times 0.30(4)$ mHz for lattice frequency detuning Δf [GHz]..." is difficult to read and might be formally inconsistent. A better way to write this would be, e.g., "transition frequency shift coefficient $\Delta\nu/\Delta f = 0.30(4)$ 10-XX U"

We agree with the reviewer about possible confusion and removed this part of the sentence from the manuscript. It does not contain any additional information.

7) Page 6, paragraph at bottom of right column: The "probability of the 806.7 nm transition" is noted with the physical unit of a rate, s^{-1} . Probably the numbers mentioned are the spontaneous decay rates of the upper level of the mentioned transitions, possibly (or not) containing a branching ratio factor. Instead of this unclear term, better refer to spontaneous decay rates or give the values of the corresponding oscillator strengths.

We are grateful to the reviewer for this remark but we believe, that in this case the terminology depends on the cited literature source. To avoid confusion, we followed traditional terminology from the ref. [Wickliffe, M. E., and Lawler, J. E. (1997). Atomic transition probabilities for Tm i and Tm ii. JOSA B, 14(4), 737-753]. The probability values given in the NIST atomic spectra database correspond to Einstein A coefficients and do not contain branching ratios.

Response to Reviewer #3

In their manuscript, Golovizin and coworkers examine the narrow $1.14\ \mu\text{m}$ optical transition in neutral thulium. Ultimately, the goal is to realize a high-performance optical lattice clock based on this transition. To the best of my knowledge, all other high-performance neutral atom optical clocks, demonstrated or currently being pursued, are based on the ground 1S_0 to metastable 3P_0 transition in alkaline earth or alkaline earth-like atoms (e.g., Sr and Yb). The clock transition considered here stands in stark contrast, which has both benefits and drawbacks. The most notable benefit is the inherent suppression of the blackbody radiation (BBR) shift, found here to be a few orders of magnitude smaller than the conventional clock transitions. This same group (at least a large overlap in authors) considered this clock transition previously in reference [34].

While that work contained experimental aspects, it was primarily a theoretical study. This work is primarily experimental, making strides towards the goal stated above. The BBR shift, the magic wavelength, and the clock frequency

are evaluated to respectable precision for a pioneering work. As best as I can tell, the experimental methods and stated uncertainties seem reasonable. As mentioned, the chosen transition has some drawbacks compared to the conventional $^1S_0 \rightarrow ^3P_0$ clock transitions. Mostly this stems from the $J \neq 0$ character of the clock states and the natural lifetime of the upper clock state. I am skeptical that the proposed clock will ever reach the performance levels already demonstrated by Sr and Yb optical lattice clocks (low- 10^{18} accuracy, sub- 10^{18} stability), but the authors do not make such a claim. All claims regarding the prospects of the clock seem well-grounded and appropriate. I believe this work is well-motivated. As correctly noted in the manuscript, the BBR shift is a major concern for

state-of-the-art optical clocks, especially the optical lattice clocks based on neutral atoms. I find the three order-of-magnitude suppression for a neutral system, as demonstrated here, to be rather astounding. As the authors point

out, this could prove useful for, say, a transportable clock. One might speculate that other similar (but narrower) transitions could be identified in other neutral atoms and perhaps tricks could be developed to circumvent, e.g., lattice polarization sensitivity (one of the drawbacks associated with $J \neq 0$). In any case, I'm intrigued. Given consideration of the comments below, I recommend publication in Nature Communications.

1) For the most part, the manuscript is well-organized and clear. My biggest complaint is grammatical errors/nuances that appear throughout (I suspect the authors are not native English speakers, which deserves some forgiveness in this regard). In the Abstract alone, I point to phrases BBR shift at the room temperature, smaller compared to, optical clocks on neutral atoms, and uncertainty in 10^{17} level.

We thank the reviewer for the comment. We asked a native speaker to check English in the manuscript.

2) One page 2 (also in Methods, Eq. (5)), where the equation $h\Delta\nu = -16a_0^3\Delta\alpha P/cw^2$ appears, it should be clearly stated that $\Delta\alpha$ in this equation is the differential polarizability in atomic units. That is, here $\Delta\alpha$ is a number that corresponds to the ratio of the differential polarizability to the atomic unit for polarizability (in one form, $4\pi\epsilon_0a_0^3$). Elsewhere in the manuscript, the symbol $\Delta\alpha$ is just the differential polarizability. For example, a couple of sentences later $\Delta\alpha$ in atomic units is intended to be read the differential polarizability in atomic units. In my opinion, the most palatable option would be to rewrite the equation as $h\Delta\nu = -16a_0^3\Delta\alpha P/cw^2$, with $\Delta\alpha$ consistently representing the differential polarizability throughout (and it can be reported in whatever units you want; Figure 2, for example, would be perfectly acceptable).

We thank the reviewer for pointing out a possible confusion with units. To avoid it, in the beginning we specified the use of atomic unit for polarizability which is then used throughout the manuscript:

The differential dynamic polarizability $\Delta\alpha$ between the clock levels was calculated using the expression $h\Delta\nu = -16a_0^3\Delta\alpha P/cw^2$. In the manuscript, we use atomic units for polarizability, namely, $1\ \text{a.u.} = 4\pi\epsilon_0a_0^3$.

3) I don't recall seeing trap depths ever given in recoil energy. This is fairly standard for this field, as it is the natural unit for motional effects in the lattice (at least in the longitudinal direction).

We agree with the reviewer that giving the trap depth in recoil energy might help readers. The following

paragraph is added to the Section “Magic wavelength determination”:

The intracavity power P was determined by calibrated photodiodes placed after the cavity outcouplers $M2$ and $M2'$. For a specified beam radius w and ground state dynamic polarizability $\alpha = 195$ a.u. at $\lambda = 813$ nm [34], the lattice depth for $P = 1$ W is $U = 82 E_r$, where $E_r = h \times 1785$ Hz is the recoil energy.

4) I did not attempt to verify equation (6) of the Methods. Still, I am surprised not to see some account of the density of motional states in the lattice (which I think would at least be apparent in the denominator). Note that the density of states is greater near the top of the trap ($E \rightarrow U_0$) than near the bottom ($E \rightarrow 0$). In line with the previous comment, knowing the depth in recoil energy would help a reader gauge whether or not a classical treatment of the motion in the trap is reasonable.

In our experiment, the minimum depth of a 1D optical lattice is about $100 E_r$, while the axial and the radial trap frequencies are $h\nu_z = 18 E_r$ and $h\nu_r = 0.02 E_r$, correspondingly. The number of axial motional states is indeed small $N_z \approx U/\nu_z \approx 5$ (more accurate simulations give 6 states), but the number of radial states is more than 3000, which justifies the validity of classical treatment. We also mentioned this in the manuscript:

Assuming a Boltzmann distribution of the atoms with temperature T in the trap of depth $U_0 \sim 100 E_r$ (whereby a classical treatment is applicable) one can calculate the parameter η connecting the averaged and maximum intensity $I_{av} = \eta I_0$:

5)The colloquial abbreviation BBR should be written out as blackbody radiation in the title.

We thank the reviewer for this comment and would like to change the title accordingly:

Inner-shell clock transition in atomic thulium with a small blackbody radiation shift

Reviewers' comments:

Reviewer #1 (Remarks to the Author):

The authors properly replied to all the comments that I made in the initial review. The manuscript is acceptable.

Here is one comment that I missed in the 1st review.

Page 5, middle:

The accuracy of the magic wavelength determination is related to the slope of $\Delta\alpha$...

I think $\Delta\alpha/\alpha$ or differential light shift at operational lattice intensity (instead of $\Delta\alpha$ alone) would be more appropriate parameter to discuss/compare the sensitivity of the magic wavelength.

Reviewer #2 (Remarks to the Author):

In my opinion, the authors' response to the referees' comments and the associated changes in the manuscript are quite adequate. I recommend publication without further changes.

Reviewer #3 (Remarks to the Author):

The authors responded adequately to all my comments, with the exception of comment #4. Even if radial degrees of freedom can be treated classically, no good argument is provided for treating the axial degree of freedom classically. In any case, I cannot make sense of equation (6). At first glance, it appears to contain the necessary ingredients for evaluating η . However, the axial degree of freedom seems to be neglected altogether, and there is no account for the density of states at different energies. Perhaps the authors could convince me, with some more effort, that equation (6) is not erroneous.

I suspect that, even if the authors were more careful with η , it would not have a major effect on the main conclusions of the paper at the stated uncertainty. In fact, even the paragraph containing equation (6) is quite opaque about what influence η had on the final determination of the magic wavelength. As it is now, I feel the discussion/treatment of the effects of finite atomic temperature is a blemish on an otherwise good paper, and I encourage the authors to improve this.

Additional minor comments:

If this manuscript proceeds to publication, the sentence introduced by the authors starting with "In the manuscript" should probably not use the word manuscript, since this word implies a to-be-published article.

On the top of page 5, there is an expression followed by "where E is the amplitude of the electric field." This is incorrect. If the expression is maintained, E should be restated as the r.m.s. electric field. For E to represent the amplitude of the electric field, an additional factor of 1/2 should be included in the expression.

Dear Reviewers,

we are thankful for your recommendation to publish our manuscript in Nature Communications, as well as for valuable comments and suggestions. We have revised our manuscript accordingly.

Please find our response to the comments below.

Response to Reviewer #1

The authors properly replied to all the comments that I made in the initial review. The manuscript is acceptable.

Here is one comment that I missed in the 1st review. Page 5, middle: The accuracy of the magic wavelength determination is related to the slope of $\Delta\alpha$

I think $\Delta\alpha/\alpha$ or differential light shift at operational lattice intensity (instead of $\Delta\alpha$ alone) would be more appropriate parameter to discuss/compare the sensitivity of the magic wavelength.

We thank the reviewer for this comment and fully agree with it. Following (Ushijima, et al., Phys. Rev. Lett., 121(26), 2018) we introduce a conventional dimensionless parameter. The rewritten text is:

*The accuracy of the magic wavelength determination is related to the slope of $\Delta\alpha(\lambda)$ in the vicinity of λ_m , which is $-0.075(17)$ a.u nm⁻¹ in Tm , as shown in the inset in Fig. 2. **In dimensionless normalized units this corresponds to $\frac{\delta(\Delta\alpha)/\alpha}{\hbar \delta\nu_L/E_r} \approx 1.5 \times 10^{-12}$ (here $\delta\nu_L$ is a lattice laser frequency increment), that is more than an order . . .***

Response to Reviewer #2

In my opinion, the authors' response to the referees' comments and the associated changes in the manuscript are quite adequate. I recommend publication without further changes.

Response to Reviewer #3

The authors responded adequately to all my comments, with the exception of comment #4. Even if radial degrees of freedom can be treated classically, no good argument is provided for treating the axial degree of freedom classically. In any case, I cannot make sense of equation (6). At first glance, it appears to contain the necessary ingredients for evaluating η . However, the axial degree of freedom seems to be neglected altogether, and there is no account for the density of states at different energies. Perhaps the authors could convince me, with some more effort, that equation (6) is not erroneous.

I suspect that, even if the authors were more careful with η , it would not have a major effect on the main conclusions of the paper at the stated uncertainty. In fact, even the paragraph containing equation

(6) is quite opaque about what influence eta had on the final determination of the magic wavelength. As it is now, I feel the discussion/treatment of the effects of finite atomic temperature is a blemish on an otherwise good paper, and I encourage the authors to improve this.

We are very grateful to the reviewer for this comment. In the revised manuscript, we introduced missing η coefficient on the top of the page 4 where we explained a $\Delta\alpha$ determination procedure, as well as in the Methods section and particularly in Eq.(5).

Following the referee note on the pronounced quantized energy levels structure in the axial direction of the optical lattice, we have extended our analysis of the average intensity to account for it. Unexpectedly, we found that for the axial vibration level with energy E the average intensity is noticeably smaller than for the radial vibrational state with the same energy, which is associated with the extension of the wave function beyond the classical turning point in a tightly confined axial direction. For example, below we list calculated coefficients for all 9 bound axial vibrational states for the trap depth $U_0 = 200E_r$, which is the typical lattice depth in our experiment: 1) energy, 2) relative occupation, 3) $\eta_z(n_z)$ coefficient (when averaging using calculated wave functions in axial direction), 4) coefficient η_{avr} , which is the average intensity for atom with corresponding energy E_{n_z} in radial direction, and 5) coefficient $\eta_r(n_z)$, which is calculated using Eq. 7 (see below):

	n_z	0	1	2	3	4	5	6	7	8
1	E_{n_z}/U_0	0.069	0.206	0.336	0.461	0.579	0.691	0.794	0.887	0.965
2	p_{n_z}	0.353	0.224	0.145	0.096	0.065	0.045	0.032	0.023	0.018
3	$\eta_z(n_z)$	0.965	0.894	0.823	0.751	0.679	0.604	0.525	0.437	0.317
4	η_{avr}	0.977	0.928	0.879	0.827	0.773	0.715	0.652	0.578	0.48
5	$\eta_r(n_z)$	0.903	0.911	0.92	0.931	0.943	0.955	0.969	0.982	0.994

Thus, estimated η coefficient is reduced from 0.90(5) to 0.76(15), as explained below and also in the paper. Stated uncertainty covers variation of η in temperature range $kT = 0.1 \dots 3U_0$, as well as model simplifications, like consideration of single-well sinusoidal potential in the axial direction and possible difference in radial and axial temperatures.

We note, that this also leads to small corrections of the differential polarizability spectra and value and uncertainty of the differential static scalar polarizability. However, the new (-0.063(30) a.u.) and previous (-0.047(18) a.u.) values of the differential static scalar polarizability are consistent within one joint standard deviation.

Page 4:

*The differential dynamic polarizability $\Delta\alpha$ between the clock levels was calculated using the expression $\mathbf{h}\Delta\nu = -16\mathbf{a}_0^3\Delta\alpha\eta\mathbf{P}/\mathbf{c}\mathbf{w}^2$. In the **paper** we use atomic units for polarizability, namely, $1\text{ a.u.} = 4\pi\epsilon_0\mathbf{a}_0^3$. Here, h is the Plank constant, c is the speed of light, \mathbf{a}_0 is the Bohr radius, and $w = 126.0(2.5)\mu\text{m}$ is the lattice beam radius as calculated from the enhancement cavity geometry. The intracavity power P was determined by calibrated photodiodes placed after the cavity outcouplers $M2$ and $M2'$. For a specified beam radius w and ground state dynamic polarizability $\alpha = 195\text{ a.u.}$ at $\lambda = 813\text{ nm}$ [34], the lattice depth for $P = 1\text{ W}$ is $U = 82E_r$, where $E_r = h \times 1785\text{ Hz}$ is the recoil energy. **Coefficient $\eta \leq 1$ accounts for the decrease of the average intensity for an atom with a non-zero kinetic energy. The details of the beam waist determination, power measurements, and coefficient η calculations are given in the Methods section....***

Methods:

The differential polarizability $\Delta\alpha$ of the clock levels is determined from the frequency shift of the corresponding transition $\Delta\nu$, circulating power P , and TEM_{00} cavity mode radius w at the atomic cloud position through

$$\Delta\alpha = -\frac{\mathbf{hcw}^2}{16\mathbf{a}_0^3\eta} \frac{\Delta\nu}{\mathbf{P}}. \quad (5)$$

*The dependency $\Delta\nu(P)$ was obtained for different wavelengths in the spectral range of 810–860 nm for intracavity circulating powers varying from 1 W to 4 W, as shown in Fig. 5. The frequency shift $\Delta\nu$ was measured relative to the laser frequency, which is stabilized to an ultrastable ULE cavity with linear drift compensation. The slope coefficients of the corresponding linear fits were substituted into Eq. (5) to deduce $\Delta\alpha$ as presented in Figs. 2 and 4. Some residual nonlinearity and, as a consequence, imperfect intersection of linear fits at $P = 0$ can be associated with finite atomic temperature **which is discussed below.***

Note that differential polarizabilities obtained from linear fits with an additional constraint $\Delta\nu(0) = 0$ agree with values obtained from our current analysis within one standard deviation.

Coefficient η accounts for the effect of the reduced average light intensity $I_{av} = \eta I_0$ due to the finite temperature of the atomic cloud in the 1D optical lattice with an intensity profile $I(\mathbf{r}, \mathbf{z}) = I_0 e^{-2r^2/w^2} \cos^2(2\pi\mathbf{z}/\lambda_m)$. Assuming a Boltzmann distribution of the atoms with a temperature T in the trap of depth $U_0 = 100 \dots 300 E_r$ (experimental lattice depth range), we estimate the coefficient η using the following expression:

$$\eta = \sum_{n_z} \frac{e^{-E_{n_z}/kT}}{Z_0} \eta_z(n_z) \eta_r(n_z). \quad (6)$$

Here the first multiplier gives the relative population of the n_z axial vibrational mode with the energy E_{n_z} (a number of axial vibrational states varies from 7 to 11 for $U_0 = 100 \dots 300 E_r$), $\eta_z(\mathbf{n}_z)$ and $\eta_r(\mathbf{n}_z)$ are the coefficients due to delocalization in the axial and the radial directions, correspondingly. To estimate $\eta_z(\mathbf{n}_z)$ we used numerically calculated wave functions of the axial vibrational modes in a single-well sinusoidal potential. $\eta_r(\mathbf{n}_z)$ was calculated using classical approximation:

$$\eta_r(n_z) = \frac{\int_0^{U_0 - E_{n_z}} e^{-E/kT} \left(\frac{1}{2r_0} \int_{-r_0}^{r_0} e^{-2r^2} dr \right) dE}{\int_0^{U_0 - E_{n_z}} e^{-E/kT} dE}, \quad (7)$$

where $r_0 = (-\ln(1 - E/U_0)/2)^{-1/2}$ is the classical turning point. We estimate parameter η to be 0.76(15) for $kT = 0.3 U_0$, which is close to an expected atomic temperature in our experiment. The stated error includes both the uncertainty of the atomic temperature as well as incompleteness of the model (the single-well sinusoidal potential instead of the optical lattice potential and a possible difference in radial and axial temperatures).

Additional minor comments:

If this manuscript proceeds to publication, the sentence introduced by the authors starting with "In the manuscript" should probably not use the word manuscript, since this word implies a to-be-published article.

We are very grateful for this comment. We've changed the word **manuscript** to **paper**.

On the top of page 5, there is an expression followed by "where E is the amplitude of the electric field." This is incorrect. If the expression is maintained, E should be restated as the r.m.s. electric field. For E to represent the amplitude of the electric field, an additional factor of 1/2 should be included in the expression.

We thank the reviewer for pointing out this inaccuracy and we corrected the statement:

In the second-order approximation, the energy shift of an atomic level $|J, F, m_F\rangle$ in an external oscillating electromagnetic field with wavelength λ equals $-\alpha_{J,F,m_F}(\lambda) \mathbf{E}^2/4$,

REVIEWERS' COMMENTS:

Reviewer #3 (Remarks to the Author):

The authors have considered my comments and made changes to the manuscript. I haven't attempted to verify the validity of the new equations (6) and (7), but at least the authors have provided sufficient detail for their approach. I'm willing to accept that, and I recommend publication. I have some additional comments below, but these are minor items. I don't feel that I need to see the manuscript again.

Other comments:

Along the same lines as a previous comment of mine, I believe equation (3) should have $E^2/4$ rather than $E^2/2$. If the equation is otherwise kept as is, then I think it should be introduced more clearly. For $\Theta \ll 1$, the clock shift attributed to the tensor polarizability additionally includes a Θ -independent term (which doesn't affect the analysis at all). What is given on the right-hand-side is the Θ -dependent shift attributed to the tensor polarizability for $\Theta \ll 1$.

The y-axis in the last figure is labeled $\Delta\nu$. Consider a different label, since $\Delta\nu$ everywhere else is associated with some systematic frequency shift.

I don't insist on this, but in the introduction the authors could sell this work from the DC Stark point of view a little more. For optical lattice clocks, there's been at least a couple of recent papers focused on this systematic effect. It's been observed to be very large in some clocks (even larger than BBR), and so it poses a legitimate concern.

Dear Reviewers,

thank you for your valuable comments. We have revised our manuscript accordingly.

Please find our response to the comments below.

Response to Reviewer #3

The authors have considered my comments and made changes to the manuscript. I haven't attempted to verify the validity of the new equations (6) and (7), but at least the authors have provided sufficient detail for their approach. I'm willing to accept that, and I recommend publication. I have some additional comments below, but these are minor items. I don't feel that I need to see the manuscript again.

Other comments:

Along the same lines as a previous comment of mine, I believe equation (3) should have $E^2/4$ rather than $E^2/2$. If the equation is otherwise kept as is, then I think it should be introduced more clearly. For $\Theta \ll 1$, the clock shift attributed to the tensor polarizability additionally includes a Θ -independent term (which doesn't affect the analysis at all). What is given on the right-hand-side is the Θ -dependent shift attributed to the tensor polarizability for $\Theta \ll 1$.

We are very grateful to the reviewer for careful reading and agree with his comments. We updated manuscript accordingly.

The y-axis in the last figure is labeled $\Delta\nu$. Consider a different label, since $\Delta\nu$ everywhere else is associated with some systematic frequency shift.

We thank the reviewer for this comment, meanwhile we would like to point out that notation $\Delta\nu$ was used throughout the manuscript for designation of the frequency shift of the clock transition frequency, both calculated and measured. However, we changed the x-label in Fig.3a, which indicates detuning of the clock laser, from $\Delta\nu$ to $\delta\nu$ to avoid possible confusion.

I don't insist on this, but in the introduction the authors could sell this work from the DC Stark point of view a little more. For optical lattice clocks, there's been at least a couple of recent papers focused on this systematic effect. It's been observed to be very large in some clocks (even larger than BBR), and so it poses a legitimate concern.

We agree with the referee that the small BBR shift is equivalent to a small sensitivity to homogeneous DC electric fields. Nevertheless, we would like to skip mentioning this in the manuscript because the clock transition might be still sensitive to spatially inhomogeneous DC electric fields due to non-zero electrical quadrupole moments of the clock levels. For these type of electric fields T_m is not so advantageous compared to other species. This discussion might confuse the reader and does not add much value to the main results of the manuscript.